# Light-Induced Charge Accumulation in PTCDI/Pentacene/Ag(111) Heterojunctions

Roberto Costantini [1] , Albano Cossaro [1,2] , Alberto Morgante [1,3] and Martina Dell'Angela [1,*]

1   CNR-IOM Laboratorio TASC, SS 14 Km 163.5, I-34149 Trieste, Italy; costantini@iom.cnr.it (R.C.);
    cossaro@iom.cnr.it (A.C.); morgante@iom.cnr.it (A.M.)
2   Department of Chemical and Pharmaceutical Sciences, University of Trieste, Via L. Giorgieri 1,
    I-34127 Trieste, Italy
3   Department of Physics, University of Trieste, Via A. Valerio 2, I-34127 Trieste, Italy
*   Correspondence: dellangela@iom.cnr.it

**Abstract:** The incorporation of singlet fission (SF) chromophores in solar cells is expected to bring significant increases in the power conversion efficiency thanks to multiexciton generation. However, efficient charge generation in the device is determined by the energy level alignment (ELA) between the active materials, which should favor exciton transport and separation under illumination. By combining ultraviolet photoemission spectroscopy and optical differential reflectance measurements, we determine the ELA in a prototypical SF heterojunction between pentacene (Pc) and perylene-tetracarboxylic-diimide (PTCDI) grown on Ag(111). Time-resolved X-ray photoelectron spectroscopy on such a system reveals light-induced modifications of the ELA; by measuring the transient shift of the core level photoemission lines we observe an accumulation of long-lived holes in the PTCDI within the first hundred picoseconds after the optical pump.

**Keywords:** organic heterojunction; pump–probe photoemission; exciton dynamics; transient energy level alignment; interface dipole





## 1. Introduction

In photovoltaic devices light is absorbed to generate excitons, bound states of an electron and a hole mutually attracted by the electrostatic Coulomb force. In order to generate photocurrent, the excitons need to be separated into free charge carriers, which are eventually collected at the electrodes. In conventional silicon-based devices, it is estimated that more than the 30% of the available solar power is lost in the thermalization of the hot excitons, which are generated by the absorption of photons with energy higher than the semiconductor band gap. To mitigate such losses, a viable strategy consists of introducing down-conversion materials in the device, thus opening a channel for the conversion of a high-energy photon into two or more low-energy excitons, and increasing the power conversion efficiency of organic solar cells [1,2]. Materials undergoing singlet fission (SF), such as polyacenes, are currently studied and employed as down-conversion materials [3–9]. However, in the realization of a SF-sensitized solar cell, the efficient carrier multiplication is not sufficient to increase the power conversion efficiency of the device and a detailed control on the transport and separation of the excitons is also necessary [7]. To ensure efficient charge, generation the active materials have to be carefully chosen, as a proper energy level alignment (ELA) across the whole device is crucial. The commonly used assumption of a constant vacuum level across the whole device that helps in material selection [10] is not always valid; in some cases, the equilibration of the chemical potential leads to the bending of the energy levels (Fermi-level pinning) [11] which causes the formation of interfacial dipoles [12–15] that may hinder charge transfer across the junction. A thorough characterization of the ELA should comprise the determination of the position of both occupied and unoccupied states; the former are typically observed via valence band

photoemission, while the latter can either be probed directly by inverse photoemission [16] or total current spectroscopy [17–19], or their position can be evaluated indirectly from the measurement of transport and optical gaps. Moreover, the effect of the optical illumination in p-n semiconductor junctions has also to been taken into account; photovoltaic effects and photoconductivity at the interfaces may transiently modify the ELA and therefore affect the device performances [20,21]. X-ray photoelectron spectroscopy (XPS) is an experimental technique sensitive to the presence of surface photovoltage effects, which cause the shift of the measured core levels [22]. The recent developments in time-resolved photoemission have opened up the possibility of measuring the dynamics of surface photovoltage from the femto- to the nano-second timescales [23]. In fact, by measuring the relaxation time of the shift of the core levels after optical excitation, it is possible to reveal transient modifications of energy levels related to the formation of excitons in molecular films [24] or to their transfer and separation at organic heterojunctions [25,26].

In this paper, we study the sub-nanosecond evolution of the ELA in an heterojunction made of pentacene (Pc) and perylene-tetracarboxylic-diimide (PTCDI) films grown on Ag(111). Both Pc and PTCDI crystals display singlet fission, and organic PV cells based on the combination of Pc and PTCDI derivatives have shown promising results in light harvesting, although with overall power conversion efficiencies in the range of 1–2% [27–29]. Here, we combine ultraviolet photoemission spectroscopy (UPS) and differential reflectance spectroscopy to determine the relative position of occupied and empty states in the system, and, by means of time-resolved X-ray photoemission (XPS), we measure the transient modifications in the ELA upon laser excitation. When pumping resonantly with PTCDI, we observe the accumulation of positive charge carriers in the PTCDI film, which determines a spectral shift in the C 1s signal persisting in the microsecond timescale. We also detect a deviation from the expected space charge shift of the C 1s peak, as compared to the Ag $3d_{5/2}$ shift, since the former is 50 meV smaller than the latter at time zero. Such effect may be caused by charge transfer processes occurring in timescales comparable to the 100 ps X-ray pulse width.

## 2. Materials and Methods

The experiments were performed at the ANCHOR-SUNDYN endstation [30] of the ALOISA beamline at the Elettra synchrotron in Trieste (Italy). The growth and characterization of the heterojunction were performed under ultra-high-vacuum (UHV) conditions. The Ag(111) single crystal was cleaned with cycles of Ar+ sputtering and annealing to 550 °C. The organic films of Pentacene (Merck KGaA, Darmstadt, Germany, Purity > 99%) and PTCDI (Alfa-Aesar, Product N. 44098) were grown via evaporation from a Knudsen cell and the growth was monitored in situ via ultraviolet photoemission spectroscopy (UPS) with He I radiation. The sample work function at all deposition steps was determined by applying a −2.3 eV bias to the sample and by measuring the secondary electron cutoff. The optical differential reflectance measurements [31] of PTCDI and Pc multilayer films on Ag(111) were measured in situ using a Hamamatsu L10290 UV-VIS fiber light source, whose beam was reflected from the sample (45° incidence) and analyzed by a S100 Solar Laser Systems spectrometer.

Time-resolved XPS measurements were performed using synchrotron pulses (~100 ps) with a photon energy of 690 eV as a probe and either 526 nm (2.36 eV) or 670 nm (1.85 eV) laser pulses (~300 fs) as a pump to selectively excite PTCDI or Pc, respectively. Given the large difference in pulse duration of the two sources, the temporal resolution is exclusively limited by the probe pulse to ~100 ps. The pump pulses are generated from the output of a Yb-doped yttrium aluminum garnet (Yb:YAG) fiber laser (Tangerine HP, Amplitude Systèmes, Pessac, France) feeding an optical parametric amplifier (Mango OPA, Amplitude Systèmes, Pessac, France). The laser repetition rate was set to 385.67 kHz and synchronized with the hybrid bunch of the Elettra synchrotron, as described in [8]. The spatial overlap of the two beams was checked on a YAG crystal mounted on the sample holder; the temporal overlap was determined by measuring the space charge shift on the Ag 3d core level. The

fluence dependence analysis was performed using the multi-bunch signal of the Elettra filling pattern, which provides an average signal over the whole acquisition window of 1.8 μs. The spectra were acquired in laser on/laser off sequences to ensure the reversibility of the effect and exclude radiation damage. The peak positions were evaluated from the fits by accounting for a rigid shift of the C 1s line after aligning the Ag $3d_{5/2}$ peak to 368.3 eV for the whole dataset.

## 3. Results

The PTCDI/Pc/Ag(111) heterojunction was grown by thermally evaporating PTCDI on a Pc multilayer on the Ag(111). Figure 1a shows UPS spectra taken at the different stages of evaporation: clean sample (gray curve), Pc multilayer on Ag(111) (cyan curve), and PTCDI/Pc/Ag(111) (magenta curve). As previously reported, when deposited on Ag(111), the Pc molecules of the first layer lay flat on the substrate surface due to the strong coupling between their π-system and the metal electrons [32]. The molecules in subsequent layers tilted due to the decrease in such interactions and followed a Stranski-Krastanov growth, with the molecules locally arranged in a herringbone-like phase [33,34]. XPS measurements show that the intensity of Ag 3d peaks did not decrease exponentially with the Pc coverage after the saturation of the C 1s signal, confirming the existence of areas of low molecular coverage and thus agreeing with the formation of molecular islands of finite size. From the comparison of our UPS spectra with literature, we can estimate an average coverage of the pentacene film between 2 and 6 nm [32]. The Pc/Ag(111) work function determined by measuring the secondary electron onset was 4.0(0.1) eV and the binding energy of the highest occupied molecular orbital (HOMO), here labelled $S_0$, of Pc molecules in their ground state was 1.7(0.1) eV. PTCDI was then deposited on top of the pentacene film. The work function of the PTCDI/Pc/Ag(111) heterojunction increased to 4.3(0.1) eV, suggesting the formation of a −0.3 eV dipole at the organic interface. We found the PTCDI $S_0$ at 2.8(0.1) eV. As we observed the almost complete suppression of the signal from the Pc $S_0$, we can conclude that the PTCDI overlayer is at least 2 nm thick, based on the inelastic mean free path of electrons with a kinetic energy of 20 eV [35]. From polarization-dependent X-ray absorption measurements on the O K-edge (not shown), we determined that the PTCDI molecules lie flat on top of the Pc, with their molecular plane almost parallel to the Ag(111) substrate. Despite the roughness of the Pc film dictated by the Stranski-Krastanov growth, the strong dichroism in the X-ray absorption spectra of PTCDI lead us to conclude that no significant mixing of the two molecular species occurs, as this would otherwise cause a more random molecular orientation. We can, therefore, locally approximate the PTCDI/Pc interface as a planar heterojunction.

Figure 1b shows the in situ optical differential reflectance spectra measured on Pc and PTCDI films. The Pc reflectance presents three maxima at 1.85, 1.95 and 2.10, eV; the first two both assigned to a transition from $S_0$ to $S_1$, the first Pc excited state [36,37]. The PTCDI spectrum in the measured range comprises a first resonance at 2.11 eV and a broader peak between 2.3 and 2.4 eV; these are the first optically allowed transitions to excited states in PTCDI films [38,39] that we label $S_{1,a}$ and $S_{1,b}$, respectively. Absorbance studies on PTCDI have shown that the growth of a band at about 2.1 eV due to the formation of PTCDI aggregates with a shrinking of the optical gap in the film with respect to the gas phase [40]. Figure 1c summarizes the measured energy level alignment in the heterojunctions. The work function values and the positions of the $S_0$ states were determined by the UPS measurements, while the relative distances between $S_1$ and $S_0$ states were set as equal to the optical gaps identified in the reflectance measurements.

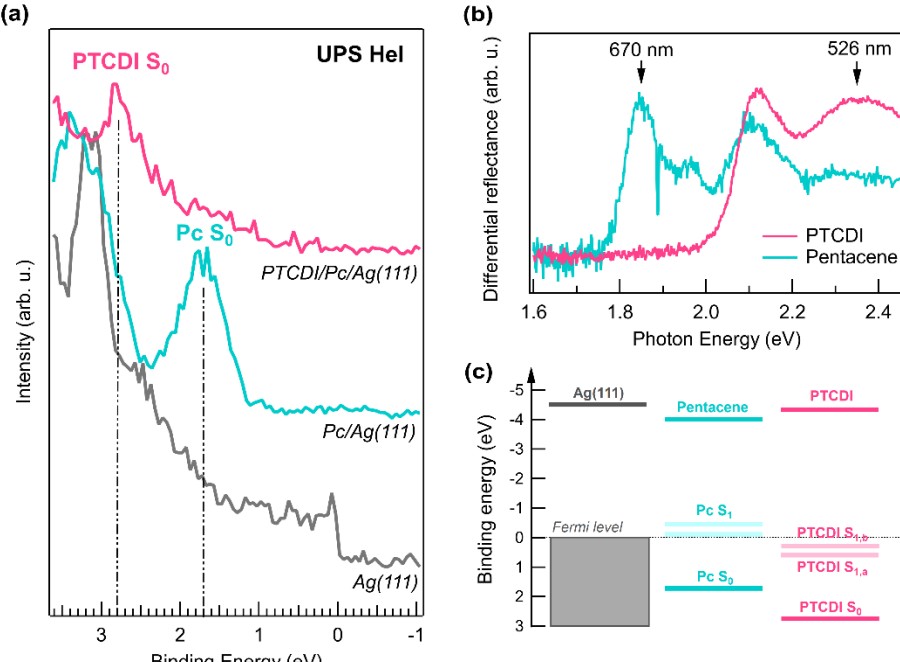

**Figure 1.** (**a**) UPS (He I) of the clean Ag(111) (gray), of Pc/Ag(111) (cyan), and of PTCDI/Pc/Ag(111) (magenta). The spectra were vertically offset for clarity. The vertical dashed lines identify the Pc $S_0$ and the PTCDI $S_0$ positions. (**b**) Optical differential reflectance spectra obtained in situ for thick films of pentacene (cyan) and PTCDI (magenta). (**c**) Scheme of the measured energy level alignment.

Figure 2 shows a selection of C 1s XPS spectra of the PTCDI/Pc/Ag(111) heterojunction upon illumination. The Pc and the main PTCDI component overlap at a binding energy of ~285 eV, while the peak at 288.5 eV corresponds to the carbon atoms in the imide groups of PTCDI [41]. The dashed gray lines are the spectra measured before illumination. The heterojunction was photoexcited by using 670 nm and 526 nm laser pulses to excite mainly Pc and PTCDI, respectively, as indicated by the arrows in Figure 1b. An initial characterization of the pump-induced effects in C 1s was performed using the multi-bunch radiation of the Elettra synchrotron as a probe, i.e., the measured shift of the core levels is averaged over the whole acquisition window, which was set to 1.8 μs. For an applied power of 10 mW, we notice that the C 1s spectrum rigidly shifts towards higher binding (lower kinetic) energies upon pumping at 526 nm (magenta curve), whereas negligible shifts (<0.05 eV) are measured if pumping at 670 nm (cyan curve). The C 1s shifts were evaluated by fitting the main photoemission component with a Gaussian peak, with an accuracy of roughly ±0.01 eV (one standard deviation). Due to the substantial overlap of Pc and PTCDI components in the main C 1s resonance, it is not possible to clearly determine whether the photoelectrons emitted from the different molecular species are equally affected by the optical excitation. However, such differences, if present, are negligible as the line shape is not visibly modified. The fluence dependence of the peak shift is reported in the inset of Figure 2: at 670 nm the shift is negligible even at 10 mW, while at 526 nm it increases logarithmically up to 0.13 eV, where it seems to plateau for a laser power above 20 mW.

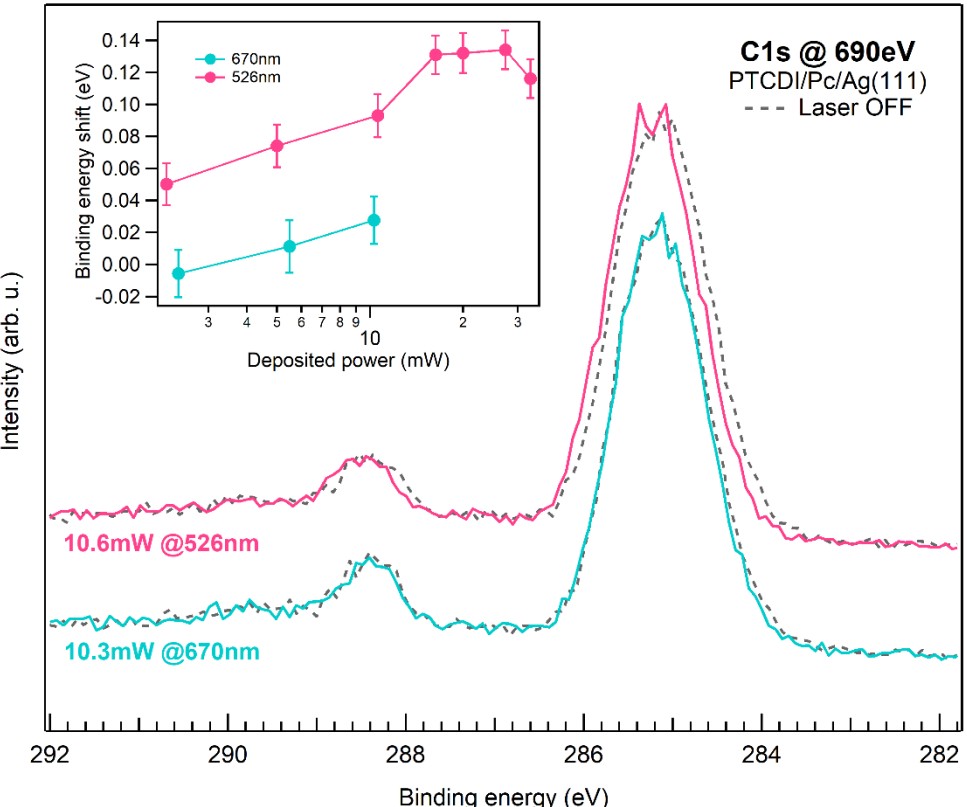

**Figure 2.** C 1s XPS spectra of PTCDI/Pc/Ag(111) measured with and without optical excitation at 526 nm and 670 nm. Upon illumination at 526 nm the C 1s core level lines shift to higher binding energy. In the inset the measured binding energy shifts as a function of deposited power for the two excitation energies are reported. Error bars represent one standard deviation.

To explore the sub-nanosecond dynamics of the C 1s core level upon photoexcitation, we also performed a 526 nm pump −690 eV probe experiment by using the hybrid mode filling of Elettra synchrotron and 100 mW optical pump power [30]. The photoemission signal was measured as a function of the time delay between the optical pump and the X-ray probe. Figure 3a shows the binding energy shifts calculated by fitting both the C 1s and Ag $3d_{5/2}$ lines in the first nanosecond after the 526 nm excitation. Both lines shift to lower binding (higher kinetic) energies in the first few hundred picosecond around time zero. This behavior is due to the vacuum space charge effect, which is caused by the interaction between the low energy electron cloud emitted by the pump and the electrons photoemitted by the probe travelling to the analyzer [30,42]. However, we notice that, in the first 50 ps, the shift in C 1s is roughly half the shift in Ag 3d. In Figure 3b, we show the C 1s spectrum acquired at time zero (solid line) after realigning the Ag $3d_{5/2}$ line to 368.3 eV to eliminate the space charge contribution, and the spectrum acquired with the second hybrid pulse after the excitation corresponded to a delay of 864 ns (dotted line) for a visual comparison. At time zero, we observe a rigid shift of the C 1s peak towards higher binding energies which is analogous to the effect described in Figure 2, but with a lifetime in the order of 50 ps. Such deviation from the space charge behavior of the C 1s with respect to the Ag $3d_{5/2}$ line suggests that charge transfer processes at the heterojunctions may also occur on sub-nanosecond timescales.

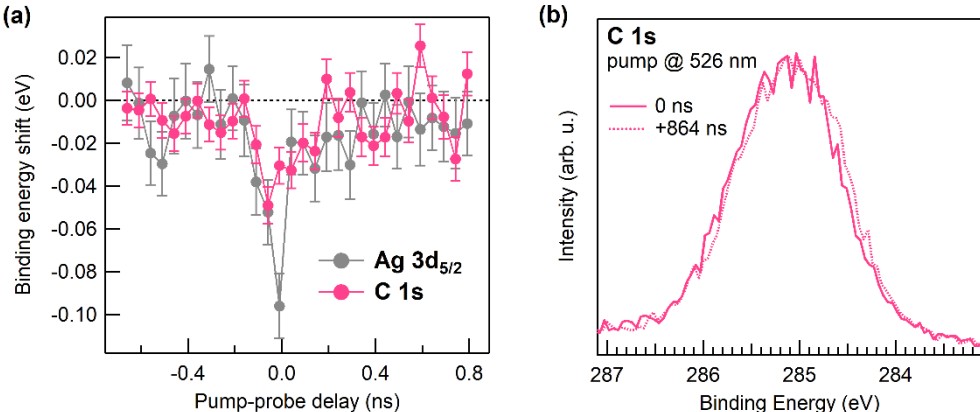

**Figure 3.** (**a**) Measured binding energy shift (Gaussian fit to the Ag 3d and C 1s lines) as a function of pump–probe delay time. Error bars represent one standard deviation. (**b**) C 1s XPS spectra measured at time zero (solid) and at 864 ns (dotted), i.e., with the second hybrid pulse after the pump, on PTCDI/Pc/Ag(111) after aligning the Ag $3d_{5/2}$ peak to 368.3 eV. The C 1s core level line shifts towards higher binding energy within the first 50 ps.

## 4. Discussion and Conclusions

We attempt now to rationalize the energy shifts of the C 1s core levels upon laser pumping shown in Figures 2 and 3 by considering the measured ELA (Figure 1c). When Pc is selectively excited at 670 nm, the photogenerated excitons can separate either at the PTCDI/Pc or at the Pc/Ag interfaces. The negligible shifts measured in the microsecond timescales (Figure 2) suggest that the latter is the more favorable decay channel and that only a small population of free charge carriers exists at long delay times. On the other hand, the 526 nm pump populates the higher level in the PTCDI $S_1$ manifold, $S_{1,b}$. Such excitons can decay to the PTCDI $S_{1,a}$ state at 2.1 eV in Figure 1b or directly separate at the PTCDI/Pc interface. The direction of the measured shifts of Figure 2 suggests that, at the microsecond timescale, the actual scenario sees a population of positive charge carriers on PTCDI, which cause the electrons photoemitted from the C 1s core level to lose up to 0.13 eV in kinetic energy. An analogous effect is found in the sub-nanosecond timescale, where we observe the C 1s peak shift to lower kinetic energies in the first 50 ps after photoexcitation. The measured direction of the charge transfer can be explained by considering that the existence of interfacial charge transfer states [43], i.e., bound pairs of electrons in Pc and holes in PTCDI, which can be directly generated by exciting the PTCDI/Pc heterojunction at 526 nm or in the decay of the $S_1$ excitons. The 0.3 eV interface dipole between Pc and PTCDI can strongly influence charge transfer across the heterojunction, representing a Coulomb barrier for the holes which may confine the positive carriers in PTCDI away from the interface, while allowing the electron transfer from Pc to PTCDI, as the barrier height is comparable to the energy difference between the $S_1$ states. Finally, both Pc and PTCDI are known to undergo singlet fission, therefore charge transfer can also occur through triplet states, which are dark state and thus cannot be probed in a static differential reflectance measurement as the one shown in Figure 1b. By considering that triplet lifetimes are of the order of 100 ps in Pc [8,44] and of 0.1–1 ns in PTCDI derivatives [45–47], we suggest that the effect measured in the first 50 ps after photoexcitation (Figure 3) can be related to the charge transfer from triplet states. Alternatively, charge carriers separated via charge transfer states at the interface can be subsequently localized in trap states in either of the two sides of the heterojunction, thus generating an induced dipole moment [48,49]. The proposed interpretations will be addressed in future two-photon photoemission measurements probing the dynamics of the excited states at the interface.

The scope of the present work is to demonstrate how pump–probe X-ray photoemission can be used in the characterization of exciton separation and charge carrier accumulation in molecular heterojunctions of interest for photovoltaic application. We have seen that, with the analysis of the kinetic energy shifts in high-resolution XPS spectra acquired

under photoexcitation, it is possible to observe the transient modifications of the energy level alignment, from the sub-nanosecond to the microsecond timescale. Such analysis may provide critical insights for the development of novel photovoltaic devices: photo-induced modifications of the ELA may hinder exciton separation and charge transfer, thus decreasing the power conversion efficiency, but can only be detected by techniques which emulate the working conditions of the device, as in the case of pump–probe spectroscopies.

**Author Contributions:** Conceptualization, R.C. and M.D.; Data curation, R.C., A.C. and M.D.; Funding acquisition, A.M. and M.D.; Investigation, R.C.; Writing—review & editing, R.C., A.C., A.M. and M.D. All authors have read and agreed to the published version of the manuscript.

**Funding:** M.D. and R.C. acknowledge support from the SIR grant SUNDYN [Nr RBSI14G7TL, CUP B82I15000910001] of the Italian Ministry of Education University and Research MIUR. The work has been supported by EUROFEL MIUR Progetti Internazionali and PRIN 2017-FERMAT (Nr. 2010KFY7XF).

**Data Availability Statement:** Data is contained within the article.

**Conflicts of Interest:** The authors declare no conflict of interest.

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
