# Peer review of "Light-Induced Charge Accumulation in PTCDI/Pentacene/Ag(111) Heterojunctions"

_chemistry, doi:10.3390/chemistry3030053_

Round 1

Reviewer 1 Report

Dell’Angela and coworkers report on ultrafast pump-probe studies using synchroton radiation (285 eV) and visible laser light (526 and 670 nm). The results are supposed to show charge carrier accumulation in pentacene or perylene-modified grown on Ag(111) heterojunction and a C 1s energy shift upon laser excitation. In my opinion, these statements are not well worked out and should be justified more convincingly.

Although the manuscript is interesting to read, it is not suitable for publication in its current form due to misleading data interpretation. Therefore, it requires major revisions (especially points 6 to 8):

  1. Abstract, first sentence: I doubt the general statement that sustainable energy sources heavily rely on organic-based light-harvesting technologies. It is definitely no result of this work and critical to be implemented in an abstract of a journal. Please give appropriate citations or change (remove) accordingly.
  2. Materials and Methods: The authors should give pulse durations both of the synchroton pulses as well as the 526 and 670 nm beams. Referencing previous publications is not helpful in this regard.
  3. Materials and Methods: Are the films sensitive to oxygen or moisture and what about long-term stability? What kind of experimental arrangements were necessary to record time-dependent traces of these films? How was sample stability controlled during and after the pump-probe experiments?
  4. Figure 1b): The laser excitation (or probing) were selected according to this differential reflectance spectrum. It would be instructive to discuss the spectral positions in films in comparison to bulk phase properties of PTDCI and pentacene.
  5. Line 156: Typo – one point too many.
  6. Figures 2 and 3 as well as Lines 172/173 (and some more): The authors should elaborate that shift more explicit because it is difficult to see. What is the accuracy of this 50 ps shift? The delay time seems to be changed by steps larger than 50 ps. What is the accuracy of the peak position of the C1s band Figure 3b)? In this figure, there is no shift within the signal-to-noise ratio; also, there is none in the 670 nm spectrum in Figure 2, but there may be one at 526 nm excitation in the same figure. Did the authors perform an error calculations and check for statistical and systematic errors? The authors show error bars in Figure 3a), but none in Figure 2 and Figure 3b). Please comment (also in the manuscript). I consider this part crucial for a final decision considering publication.
  7. Line 193: The authors claim to have observed a shift of 0.13 eV at an absolute energy of 285 eV. This would correspond to an accuracy of better than 1/2000 with the current technique. I doubt this accuracy. Please give evidence.
  8. Line 208: Why is this shift interpreted as charge transfer from triplet states? What is the evidence for triplet state formation and what is known about the timescale of intersystem crossing in these films? What other possibilities would support the data equally well? What about formation of trapped states, charge separation etc.? Please provide appropriate references in the discussion section.

Reviewer 2 Report

The manuscript of ‘Light-induced charge accumulation in PTCDI/pentacene/Ag(111) heterojunctions’ authored by Costantini et al. demonstrates the characterization of exciton separation and charge carrier accumulation in PTCDI/pentacene/Ag(111) heterojunctions by employing pump-probe XPS technique. The authors provide the analysis of the kinetic energy shifts in high-resolution XPS spectra, offering significant hints for the development of novel organic solar cells. The overall writing is good and clear, and I recommend publishing this work in Chemistry after the authors consider below minor comments.

 A couple of comments:

  1. There are some typos and grammar errors in the content, which are required to be rectified. For instance, page 4 line 141, seet should be set.
  2. What’s the instrument response file (IRF) or time resolution for the employed time-resolved XPS system? Please clarify.
  3. Some basic properties of PTCDI/pentacene/Ag(111) heterojunctions are recommended to be presented, such as optical and structural properties.

The authors used vapor method to prepare the organic films. How about the solution process which is low-cost and suitable for large-scale manufacturing? The authors should give some discussion on the comparison of different methods to make singlet fission materials by referring some other advanced studies, i.e. https://onlinelibrary.wiley.com/doi/full/10.1002/aenm.201801720, https://pubs.acs.org/doi/abs/10.1021/acs.jpca.0c01791.

Reviewer 3 Report

The paper reports on novel and valuable results on characterization of exciton separation and charge carrier accumulation in molecular heterojunctions using pump-probe X-ray photoemission. The authors carried out a very versatile analysis of the kinetic energy shifts in high-resolution XPS spectra acquired under photoexcitation, which might be used for determining energy level alignment at organic/organic material and organic/inorganic material interfaces.

As to reviewer comments, questions and/or suggestions there is as follows.

1. In the body of the paper including Fig.1 the authors present a large portion of results obtained using UPS and differential reflectance spectroscopy for determining the relative position of occupied and empty states. Those are very valuable results, so it would be useful to mention the techniques and the results also in the abstract section.

2. The paper would strongly benefit if the authors broaden the discussion (or introduction section) on the phenomenon of energy level alignment and the interface charge transfer. Particularly, short- and long(extended) – interface dipole formation, Fermi level pinning and diffusion of atomic components over the interface are the subject of interest for a potential reader. A lot of such results are obtained as a result of investigations of occupied and empty states using UPS and Total Current Spectroscopy (TCS) the approach which correspond well to the part of the present work mentioned above in the item 1. So I recommend that a few lines on the subject of the interfacial interaction are to be added into the introduction and/or discussion section. Below are some refs on the subject which I found as examples.

1. Gruenewald, L.K. Schirra, P. Winget, M. Kozlik, P.F. Ndione, A.K. Sigdel, J.J. Berry, R. Forker, J.-L. Brédas, T. Fritz, O.L.A. Monti, Integer charge transfer and hybridization at an organic semiconductor/conductive oxide interface, J. Phys. Chem. C 119 (2015) 4865.

  1. L. Shu, W.E. McClain, J. Schwartz, A. Kahn, Interface Dipole Engineering at Buried Organic–Organic Semiconductor Heterojunctions, Organic Electron. 15 (2014) 2360.
  2. A.S. Komolov, E.F. Lazneva, N.B.Gerasimova, Y.A. Panina, A.V. Baramygin, G.D. Zashikhin, .A. Pshenichnyuk, Structure of vacant electronic states of an oxidized germanium surface upon deposition of perylene tetracarboxylic dianhydride films, Physics of the Solid State 58, (2016) 377.
  3. A.S. Komolov, E.F. Lazneva, N.B.Gerasimova, Y.A. Panina, V.S. Sobolev, A.V. Koroleva, S.A. Pshenichnyuk, N.L. Asfandiarov, A. Modelli, B. Handke, O.V. Borshchev, S.A. Ponomarenko, Conduction band electronic states of ultrathin layers of thiophene/phenylene co-oligomers on an oxidized silicon surface, J. Electron Spectr. Rel. 235 (2019) 40.
  4. A.S. Komolov, E.F. Lazneva, S.N. Akhremtchik, N.S. Chepilko, A.A. Gavrikov, Unoccupied Electronic States at the Interface of Oligo(phenylene-vinylene) Films with Oxidized Silicon, J. Phys. Chem. C 117 (2013) 12633.

I recommend this manuscript for publication after minor revision.  

Regards, Reviewer---------

-----------------------------

Round 2

Reviewer 1 Report

The authors present a revised version and answered all questions of the referees satisfactorily. Apart from answering all points they also changed the manuscript accordingly. I recommend publication, maybe with a small change considering the last point (8) regarding triplet states vs. charge transfer:

In line 235, it reads "... thus generating a dipole ...". Is this a permanent dipole or should it better read "... thus generating an induced dipole moment"?